# Oxidative Stress in Patients with Advanced CKD and Renal Replacement Therapy: The Key Role of Peripheral Blood Leukocytes

**DOI:** 10.3390/antiox10071155

**Published:** 2021-07-20

**Authors:** Carmen Vida, Carlos Oliva, Claudia Yuste, Noemí Ceprián, Paula Jara Caro, Gemma Valera, Ignacio González de Pablos, Enrique Morales, Julia Carracedo

**Affiliations:** 1Department of Genetics, Physiology and Microbiology (Unit of Animal Physiology), Faculty of Biology, University Complutense of Madrid, 28040 Madrid, Spain; coliva02@ucm.es (C.O.); nceprian@ucm.es (N.C.); gvalera@ucm.es (G.V.); 2Research Institute Hospital 12 de Octubre (imas12), 28041 Madrid, Spain; claudia.yuste@salud.madrid.org (C.Y.); paulajara.caro@salud.madrid.org (P.J.C.); ignacioangel.gonzalez@salud.madrid.org (I.G.d.P.); emoralesr@senefro.org (E.M.); 3Department of Nephrology, Hospital Universitario 12 de Octubre, 28041 Madrid, Spain

**Keywords:** chronic kidney disease, oxidative stress, polymorphonuclear leukocytes, mononuclear leukocytes, hemodialysis, peritoneal dialysis

## Abstract

Oxidative stress plays a key role in the pathophysiology of chronic kidney disease (CKD). Most studies have investigated peripheral redox state focus on plasma, but not in different immune cells. Our study analyzed several redox state markers in plasma and isolated peripheral polymorphonuclear (PMNs) and mononuclear (MNs) leukocytes from advanced-CKD patients, also evaluating differences of hemodialysis (HD) and peritoneal dialysis (PD) procedures. Antioxidant (superoxide dismutase (SOD), catalase (CAT), glutathione peroxidase (GPx), reduced glutathione (GSH)) and oxidant parameters (xanthine oxidase (XO), oxidized glutathione (GSSG), malondialdehyde (MDA)) were assessed in plasma, PMNs and MNs from non-dialysis-dependent-CKD (NDD-CKD), HD and PD patients and healthy controls. Increased oxidative stress and damage were observed in plasma, PMNs and MNs from NDD-CKD, HD and PD patients (increased XO, GSSG and MDA; decreased SOD, CAT, GPX and GSH; altered GSSG/GSH balance). Several oxidative alterations were more exacerbated in PMNs, whereas others were only observed in MNs. Dialysis procedures had a positive effect on preserving the GSSG/GSH balance in PMNs. Interestingly, PD patients showed greater oxidative stress than HD patients, especially in MNs. The assessment of redox state parameters in PMNs and MNs could have potential use as biomarkers of the CKD progression.

## 1. Introduction 

Chronic kidney disease (CKD) is a high global systemic pathology that affects approximately between 11–13% of the population worldwide [1], with the most prevalence in elderly people [2]. Indeed, due to the aging population, CKD is exponentially increasing and is the main contributor to morbi-mortality from non-communicable diseases [3]. In this regard, several studies have shown that patients with CKD suffer accelerated aging [4,5], contributing to increased risk of adverse outcomes, such as cardiovascular diseases (CVDs), altered immune response, neurological complications, or premature death [5,6,7]. Finally, CKD may advance to end-stage renal disease (ESRD) requiring renal replacement therapy (RRT), such as hemodialysis (HD) or peritoneal dialysis (PD) [6,8].

A chronic oxidative stress situation (a progressive redox imbalance due to the excessive production and accumulation of oxidant compounds in cells and tissues that exceeds the scavenging capacity of antioxidant systems to detoxify these reactive products), together with a low grade of inflammation, are the basis of aging and several age-related diseases [9,10,11]. Thus, in addition to traditional risk factors (e.g., hypertension, diabetes mellitus, obesity, etc.), both chronic systemic inflammation and excessive oxidative stress also contribute to the onset, progression and pathogenesis of CKD and renal failure [5,12,13,14,15,16], being some of the major contributors to the associated-CVDs and elevated mortality in CKD patients [5,8]. It is also important to highlight that oxidative-inflammatory stress is increased even in the early stages of CKD and progresses in parallel to the deterioration of renal functions [5,12,14,15]. Moreover, the presence of other non-conventional risk factors, such as anemia [17,18], or several lifestyle factors, such as nutrition or dietary interventions [6,19], also have a great impact on the redox status of CKD patients, promoting oxidative-inflammatory stress, and consequently, aggravating progression to renal failure and CVDs in these patients [6,17,18,19]. In fact, advanced CKD patients may suffer from malnutrition or dietary restrictions, which, together with the loss of micronutrients during dialysis procedures, may further increase oxidative stress due to the reduction of extracellular antioxidant defenses [12,13,14]. In addition, an inadequate intake of oligoelements (e.g., copper, manganese, etc.) may draw on misleading antioxidant enzymes [19]. However, medical nutrition therapy (e.g., low-protein diet restriction) may be useful in decreasing uremic toxicity and increasing the efficiency of muscle and body metabolism in severe CKD patients [20].

Enhanced oxidative-inflammatory stress is not restricted to the kidney [8] but is also present in other peripheral locations, such as plasma or immune cells [21]. Thus, at peripheral levels, CKD patients exhibit increased oxidative stress and a chronic inflammation state [5,12,13,14,15], as well as an impaired immune response with/or activation of peripheral blood polymorphonuclear (PMNs) and mononuclear (MNs) leukocytes in the circulation [6,15,16,22,23]. Indeed, the leukocyte activation promotes an increased release of pro-inflammatory cytokines, such as interleukin-(IL)-6 or IL-1β, as well as overproduction of oxidant compounds, such as reactive oxygen species (ROS) [5,23]. Thus, the overproduction of ROS by the increased activity of several oxidant enzymatic systems (e.g., xanthine oxidase (XO), myeloperoxidase, etc.) or by the dysregulation of the mitochondrial respiratory chain, together with the impairment of the antioxidant systems, including enzymes (e.g., superoxide dismutase (SOD), catalase (CAT), glutathione peroxidase (GPx), etc.) and other non-enzymatic compounds (e.g., reduced glutathione (GSH), etc.) systems [5,24,25,26,27], further leading to oxidation of macromolecules (e.g., lipids, proteins and DNA) and, consequently, cellular and tissue damage [5,15,16,28]. Indeed, CKD patients show elevated levels of oxidized compounds (e.g., oxidized glutathione (GSSG), etc.), lipid peroxidation (e.g., malondialdehyde (MDA), etc.) and oxidized proteins and DNA [23,29], contributing to dysfunction in CKD patients [5,30] and especially in HD and PD patients [12,13,14,15,28,31,32]. Additionally, since oxidation and inflammation are interlinked processes [9], the accumulation of oxidant compounds triggers the activation of inflammatory mediators and, subsequently, amplify and perpetuate a vicious circle of oxidative damage [23], contributing to chronic kidney damage and systemic complications in CKD, such as CVDs [5,15,22,28]. However, most studies have been carried out on plasma, serum or monocytes of CKD patients but not in other types of peripheral blood leukocytes. In fact, human studies about the changes in redox state parameters in different types of leukocytes and at different stages of CKD are still scarce and preliminary.

The CKD patients undergoing RRT also exhibit higher levels of inflammation and oxidative stress compared with non-dialysis-dependent CKD patients (NDD-CKD) [12,13,14]. In fact, both HD and PD procedures are characterized by amplifying oxidative-inflammatory stress in dialysis patients by different underlying mechanisms [5,12,28,31,32]. On the one hand, the HD technique aggravates the accumulation of oxidative products due to several factors, such as duration of dialysis, the filtration of low molecular weight antioxidants, use of anticoagulants (e.g., heparin, etc.), or both the biocompatibility of dialyzer membrane (e.g., polisulfone, cellulose, etc.) and the type of dialysates (e.g., citrate, acetate, etc.) [8,12,14,33,34,35]. In fact, it is known that acetate dialysates induce complement and leukocytes activation, enhancing ROS production and the release of pro-inflammatory cytokines during HD, whereas citrate dialysates increase antioxidant enzymes (e.g., SOD, GSH) and reduce oxidized compounds and lipid damage (e.g., MDA, GSSG, etc.) [34,35,36]. Moreover, a few studies have also shown different patterns in the redox state and inflammation of HD patients depending on the type of dialyzer membrane (e.g., polisulfone, cellulose, etc.) and the HD modality (e.g., low- and high-flux HD, hemodiafiltration, etc.) [37,38,39,40]. However, these reports are not clear, showing increased, decreased or no changes in the oxidative-inflammatory stress parameters analyzed [37,38,39,40]. On the other hand, the composition of the PD dialysate solutions, together with chronic exposure of the peritoneal membranes, can trigger oxidative stress and inflammation in PD patients [12,13,14,32]. Thus, the increased glucose and lactate concentrations, low pH or hyperosmolality of the PD dialysates can promote ROS overproduction and the accumulation of oxidative damage products in the peritoneum, enhancing calcification and fibrosis [12]. Regarding PD modality, it is known that continuous ambulatory peritoneal dialysis (CAPD) also enhances both oxidative stress and lipid damage in plasma and red blood cells (e.g., MDA, GSSG, etc.) from PD patients compared with healthy subjects and NDD-CKD patients [12,13,41]. In addition, several studies also showed a strong linkage between residual renal function (RRF) and oxidative stress, correlating with CVDs and survival in PD patients [12]. Thus, a preserved RRF is associated with reduced oxidative stress and lipid damage in PD patients [12,42,43]. Nevertheless, the PD is considered a more biocompatible dialysis technique compared to the HD [12]. Although several studies have found a higher accumulation of oxidants compound and antioxidant depletion in HD patients compared with PD patients [12,13,32,44], the data are still scarce. Moreover, most research evidence has been evaluated in plasma/serum analysis but not in other blood cells from CKD patients. In this context, it is important to note that numerous peripheral blood cell types are involved in the maintenance of the redox state [10]. However, the most studied and commonly used in the clinical setting are MNs (mainly lymphocytes) but not other cells, such as isolated PMNs (phagocytes: mainly neutrophils). Since few studies have been suggested that phagocytes are the principal cells behind the chronic oxidative stress and damage associated with aging and age-related diseases [9,10,11], these cells could provide a helpful sample to clarify the molecular mechanisms underlying the increased oxidative stress throughout CKD progression. Furthermore, it is possible that the assessment of several markers of redox state in PMNs and MNs leukocytes may be useful biomarkers of the CKD progression. However, to our knowledge, no studies on this issue have been explored in HD or PD patients.

Therefore, the aim of this study was to analyze the changes in several oxidative stress and damage parameters in different types of peripheral blood leukocytes from advanced CKD patients (stages 4–5) and relative age-matched healthy controls. In addition, differences by RRT were also considered. Therefore, CKD patients were subdivided into NDD-CKD, HD and PD patients. To address this study, several pro-oxidant enzymes (XO) and oxidized compounds (GSSG), antioxidants enzyme activities and compounds (SOD, CAT, GPx and GSH), lipid oxidative damage (MDA) and inflammatory markers were investigated in both plasma and/or isolated PMNs and MNs cells from NDD-CKD, HD and PD patients and healthy subjects.

## 2. Materials and Methods

### 2.1. Research Participants

For this cross-sectional study, a total of 117 volunteers were finally enrolled selected and divided into four experimental groups: healthy controls (*n* = 17), advanced NDD-CKD (*n* = 39), HD (*n* = 39) and PD (*n* = 22) patients. All the patients were selected from a cohort of prevalent patients from the Nephrology Department, Hospital Universitario 12 de Octubre (Madrid, Spain). The study determinations were performed between February 2018 and December 2019. All procedures were performed according to good clinical practice guidelines, and all patients gave their written informed consent statements for study participation [45]. The study protocol was approved by the ethics in human research committee of Hospital Universitario 12 de Octubre (Ethical approval code: CEIC 17/407). NDD-CKD patients were selected by disease severity and impaired renal function (stages 4 and 5, estimated glomerular filtration rate, eGFR < 30 mL/min per 1.73 m^2^). Regarding RRT, all the patients undergoing HD and PD in the Nephrology Department of Hospital Universitario 12 de Octubre. The only exclusion criterion was an onset of chronic HD or PD shorter than 30 days before the study. On the one hand, the prescription of HD sessions was based on a regimen of 3 sessions per week (4 h each). Pre-dialysis blood samples for measurement were drawn on mid-week treatment (Wednesday or Thursday). HD patients received online hemodiafiltration (75%) or conventional high-flux HD (25%) with polysulfone, asymmetric cellulose triacetate (CTA) and polyacrylonitrile (AN69) membrane dialyzers (95%, 2.5% and 2.5%, respectively) (Helixone^®^ membrane, ©Fresenius Medical Care, Bad Homburg, Germany). Standard ultrafiltered citrate buffer (SelectBag Citrate^®^, Baxter, Madrid, Spain) was used as dialysate. Moreover, HD patients (90%) received heparin-based anticoagulants (95% sodium heparin and 5% low molecular weight). On the other hand, PD patients received automated (APD) or continuous ambulatory (CAPD) peritoneal dialysis (86% and 14%, respectively); citrate dialysates containing 1.5–2.5% glucose solution (©Fresenius Medical Care; 41% of PD patients) or 1.36% to 2.27% icodextrin content (Baxter; 59% of PD patients) were used. Regarding RRF, 16 HD patients (41%) had residual diuresis between 100–1300 mL/day, while 20 PD patients (91%) had RRF between 500–3000 mL/day.

Clinical and analytical variables selected for the study were recorded at the time of extraction of blood samples. These included age, gender, CKD etiology and comorbidities (arterial hypertension, hyperuricemia, diabetes mellitus, dyslipidemia, CVDs and tumors). Previous immunosuppression, treatment with erythropoietin stimulating agents or allopurinol and the type of vascular access were also recorded. The following analytical parameters were also analyzed in serum/plasma: albumin, creatinine, total proteins, hemoglobin (Hb), glycosylated hemoglobin (HbA1c), total cholesterol (TC), total triglycerides (TG), uric acid (UA) and C-reactive protein (CRP).

### 2.2. Collection of Blood Samples and Processing

Peripheral blood samples were collected using vein puncture and EDTA-buffered Vacutainer tubes (BD Diagnostic, Madrid, Spain). Blood extraction was performed between 9:00 to 10:00 a.m. to avoid circadian variations and after an overnight fast of 12 h. On the one hand, plasmas were obtained for biochemical analysis according to standard protocols. Plasma concentrations of albumin, creatinine, total proteins, Hb, TC, TG, HbA1c, UA and CRP were measured with an automated clinical biochemistry analyzer (Hitachi 7050, Hitachi Corp., Tokyo, Japan) and a hematology analyzer (KX21N, Sysmex, Kobe, Japan). On the other hand, whole blood (12 mL) was used for isolation of PMNs and MNs leukocytes following a previously described method [11]. Thus, PMNs and MNs cells were isolated using 1.119 and 1.077 density Hystopaque (Sigma-Aldrich, Madrid, Spain) separation, respectively. Collected cells were counted (95% of viability determined using trypan blue staining test) and adjusted to a specific number of leukocytes, depending on the parameter analyzed. For that, PMNs and MNs leukocytes were washed with cold PBS 0.05 M, pH 7.4 and centrifuged at 1200× *g* for 10 min at 4 °C, the supernatants were removed, and the aliquots of the pellet were stored at −80 °C until use. Moreover, plasma samples were also stored at −80 °C until use for the determination of inflammatory and oxidative stress parameters.

### 2.3. Peripheral Blood Lymphocytes Population Subtyping 

Peripheral blood lymphocyte populations (PBLP): total lymphocytes and the following lymphocyte subtyping: CD3+ T-cell, CD4+ T-cell, CD8+ T-cell, CD19+ B-cells and CD3−CD56+CD16+ NK cells, were assessed following a previously described method [46]. Determination of PBLP was performed with a FACSCanto II flow cytometer, and the data were analyzed by FACSCanto clinical software (BD Biosciences, San Jose, CA, USA). Values were compared with the range of normality established by our laboratory using blood samples from healthy controls.

### 2.4. Superoxide Dismutase Activity

The total SOD activity was measured using a colorimetric assay kit (EnzyChromTM ESOD-100, BioAssay Systems, Hayward, CA, USA). In the assay, superoxide (O_2_^•−^) is provided by XO catalysed reaction. O_2_^•−^ reacts with a water-soluble tetrazolium (WST-1) dye to form a coloured product. SOD scavenges the O_2_^•−^. Thus, less O_2_^•−^ is available for the chromogenic reaction. Plasmas were diluted 1:5 according to the manufacturer’s instructions. PMNs and MNs leukocytes (1 × 10^6^ cells) were resuspended in cold lysis buffer (500 µL), sonicated and centrifuged (12,000× *g*, 5 min; 4 °C). Clear supernatants and plasmas (20 µL) were incubated with XO and WST-1 for 1 h, and absorbance was measured at 440 nm. The results were expressed as units (U) SOD/mg protein or U SOD/mL.

### 2.5. Catalase Activity 

The CAT activity was determined using a fluorimetric kit (A-22180 Amplex^®^ Red Catalase Assay Kit, Molecular Probes, Paisley, UK). In the assay, CAT reacts with hydrogen peroxide (H_2_O_2_) to produce water and oxygen. The Amplex Red reagent containing horseradish peroxidase (HRP) reacts with any unreacted H_2_O_2_ (1:1 stoichiometry) to produce the fluorescent resorufin. Plasmas were assayed directly according to the manufacturer´s protocol. PMNs and MNs leukocytes (1 × 10^6^ cells) were resuspended in lysis solution (100 µL), sonicated and centrifuged (3200× *g*, 20 min; 4 °C). Clear supernatants and plasmas (25 μL) were mixed with H_2_O_2_ 40 μM for 30 min at room temperature. After incubation (30 min at 37 °C) with the working solution, fluorescence was measured at 530–590 nm excitation-emission detection. The results were expressed as U CAT/mg protein or U/mL.

### 2.6. Gluthathione Peroxidase Activity 

The GPx activity was assessed using a colorimetric assay kit (EnzyChromTM EGPX-100, BioAssay Systems, USA), which quantifies the consumption of nicotinamide adenine dinucleotide phosphate (NADPH) in the GPx-coupled reactions by the decline in absorbance at 340 nm (directly proportional to the GPx activity). Plasmas were assayed directly according to manufacturer´s instructions. PMNs and MNs leukocytes (1 × 10^6^ cells) were resuspended in cold lysis buffer, sonicated and centrifuged (14,000× *g*, 10 min; 4 °C). Clear supernatants and plasmas (10 μL) were mixed with the working reagent assay buffer and substrate solution. Absorbance was measured at 340 nm immediately (time zero) and again at 4 min. The results were expressed as U GPx/mg protein or U/L. 

### 2.7. Xanthine Oxidase Activity

The XO activity was quantified using a commercial kit (A-22182 Amplex Red Xanthine/Xanthine Oxidase Assay Kit, Molecular Probes). In the assay, XO catalyzes the oxidation of xanthine/hypoxanthine to UA and O_2_^•−^, which spontaneously degrades to H_2_O_2_. Amplex Red reagent containing HRP reacts with H_2_O_2_ to generate the resorufin. The XO assay was evaluated in plasma (50 μL) and isolated PMNs and MNs leukocytes (3 × 10^6^ cells) as previously described [10]. Red-fluorescent resofurin production was measured at 530–590 nm excitation-emission detection. Results were expressed as milliunits (mU) XO/mg protein or mU/mL.

### 2.8. Glutathione Content Assay

Both reduced (GSH) and oxidized (GSSG) glutathione were quantified using a fluorimetric assay [47], which is based on the reaction of GSSG and GSH with o-phthalaldehyde (OPT, Sigma-Aldrich, Spain), at pH 12 and pH 8, respectively, resulting in the formation of a fluorescent product measured at 420 nm. The assay was evaluated in plasma (50 μL) and isolated PMNs and MNs leukocytes (3 × 10^6^ cells) as previously described [11]. Results were expressed as nmol/mg protein or nmol/mL. Furthermore, the GSSG/GSH coefficient was also calculated.

### 2.9. Lipid Peroxidation Assay

Malondialdehyde (MDA) content, the most commonly used maker of lipid peroxidation, were assessed using the commercial kit “MDA Assay Kit” (Biovision, Milpitas, CA, USA), which measures the reaction of MDA with thiobarbituric acid (TBA) and the MDA-TBA adduct formation measured at 532 nm. The assay was evaluated in plasma (300 μL) and isolated PMNs and MNs leukocytes (1 × 10^6^ cells) as previously described [11]. Results were expressed as nmol MDA/mg protein or MDA/mL.

### 2.10. Protein Content Assay

The protein contents of PMNs and MNs were determined using the bicinchoninic acid protein (BCA) assay kit protocol (Sigma-Aldrich, Madrid, Spain) according to the manufacturer´s instructions. 

### 2.11. Cytokine IL-1β Measurement

The basal release of IL-1β was measured in plasma using a multiplex luminometry detection system (MILLIPLEX human high sensitivity T cell magnetic bead panel, HSTCMAG-28SK, EMD Millipore, MA, USA), with minimum detectable doses of IL-1β under 0.14 pg/mL. The results were expressed as pg/mL.

### 2.12. Statistical Analysis 

Statistical analysis was performed in SPSS Statistics 21.0 (IBM, Chicago, IL, USA). All tests were two-tailed, with a significant level of *α* = 0.05. Data were presented with a percentage and mean ± standard deviation (SD) for categorical and continuous variables, respectively. To compare the clinical features and comorbidities of the different groups, Pearson χ^2^ test and one-way analysis of variance (ANOVA) were used. Normality and homogeneity of the variances were tested by the Kolmogorov-Smirnov test and Levene test, respectively. To determine the differences between groups, we used the Student´s t-test for independent samples as well as one-way ANOVA followed by non-parametric Kruskal–Wallis test or post hoc comparison analysis (Tukey test and Games–Howell test for homogeneous or unequal variances, respectively). Linear correlations between biochemical, inflammatory and oxidative stress parameters were explored using bivariate Pearson´s correlation coefficients (*r*). GraphPad Prism 6 Software (LLC, San Diego, CA, USA) was used to perform the figures. 

## 3. Results

### 3.1. Demographical, Clinical and Biochemical Characteristics

The demographic and clinical characteristics of the participants are shown in Table 1. No differences in the mean age were found between the different study groups. A similar percentage of males and females was observed in the control and HD groups, whereas more male than female patients were included in NDD-CKD and HD groups.

Regarding clinical comorbidities (Table 1), 35 NDD-CKD (89.7%), 33 HD (84.6%) and 20 PD (90.9%) patients had hypertension, whereas 17 NDD-CKD (43.6%), 7 HD (17.9%) and 5 (22.7%) PD patients had diabetes mellitus. Almost half of the NDD-CKD, HD and DP patients had dyslipidemia, and the other half had hyperuricemia and various CVDs. HD and PD had a lower % of diabetes, dyslipidemia and hyperuricemia than NDD-CDK patients, whereas PD patients showed a higher % of diabetes and hyperuricemia and lower % of peripheral vascular disease than HD patients (see *p*-values in Table 1). CKD etiologies were similar among the HD and PD groups, with glomerulonephritis being the predominant cause of renal disease (28% and 36%, respectively), whereas in NDD-CKD patients, the main cause was diabetes (31%). Furthermore, almost half of the NDD-CKD, HD and DP patients used medications (e.g., antiplatelet agents, statins, allopurinol and erythropoietin). NDD-CKD group included significantly more patients receiving statins and less patients being treated with antiplatelet compared to dialysis patients. HD group had a significantly higher and lower % of patients receiving erythropoietin and allopurinol, respectively, than the NDD-CKD and PD groups. Regarding RRT, 56.41% of HD patients had an AVF as vascular access. In PD, 75% and 25% of PD patients underwent APD and CAPD, respectively. The RRF was higher in the PD group (*p* < 0.01) than in the HD group.

Biochemical characteristics, inflammatory markers and lymphocytes populations of the study groups are also displayed in Table 2. Regarding biochemical variables, the NDD-CKD, HD and PD patients showed lower albumin and Hb content and higher serum creatinine (*p* < 0.001) levels than healthy controls. Moreover, NDD-CKD patients had also higher TG (*p* < 0.01) and UA (*p* < 0.05) levels than controls. Interestingly, dialysis patients had lower TG (*p* < 0.01 in HD; *p* < 0.05 in PD) and higher serum creatinine (*p* < 0.001) levels than NDD-CKD patients. Interestingly, HD patients had lower TC and HbA1c contents (*p* < 0.05) than NDD-CKD patients, whereas the PD group showed lower albumin content (*p* < 0.05) than the NDD-CKD group. No significant differences were observed for the other parameters.

Regarding the inflammatory markers, we assessed plasma CRP levels, together with extracellular basal IL-1β release under resting conditions (an indicator of sterile inflammation). NDD-CKD patients showed higher basal IL-1β and CRP levels (*p* < 0.05) than healthy subjects. In addition, HD and PD patients also had higher CRP levels than control (*p* < 0.01) and NDD-CKD (*p* < 0.05) groups. Regarding RRT, no differences were observed in IL-1β and CRP levels between HD and PD groups.

We also investigated the changes in lymphocytes populations, measuring the surface-marker profile by flow cytometry (Table 2). The lymphocytes subsets were divided between T-cells (CD3+; isotype CD4+ or CD8+), B-cells and NK cells. No significant differences were observed between NDD-CKD and control groups. However, dialysis patients showed a significant decrease in the total number of lymphocytes, as well as a lower number of CD3+ and CD4+ T-lymphocytes, B-lymphocytes and NK cells, and the CD4+/CD8+ ratio than controls (see *p*-values in Table 2). Interestingly, HD patients had a marked reduction in the total number of lymphocytes, CD3+ and CD4+ T-lymphocytes and NK cells, as well as a lower CD4+/CD8+ ratio than NDD-CKD patients. Moreover, PD patients had a lower total number of lymphocytes, B-cells and CD4+/CD8+ compared with NDD-CKD patients (see *p*-values in Table 1). No significant differences between HD and PD-group were found.

### 3.2. Changes in Oxidative Stress and Lipid Damage Parameters in Plasma

The results regarding oxidative stress and lipid damage parameters in plasma from healthy subjects and NDD-CKD, HD and PD patients are shown in Figure 1.

The pro-oxidant XO activity (Figure 1A) was significantly increased in NDD-CKD *(p* < 0.05*)* and PD (*p* < 0.01) patients in comparison with healthy controls. Moreover, HD and PD patients showed lower and higher XO activity (*p* < 0.05), respectively, compared with NDD-CKD patients. Interestingly, PD patients had significantly higher XO activity (*p* < 0.01) than HD patients.

Furthermore, we also observed impairment of the circulating antioxidant enzymes SOD, CAT and GPx (Figure 1B–D), as well as of the GSH content (Figure 1E). In fact, NDD-CKD, HD and PD patients showed lower CAT and GPx activities and GSH concentration*,* as well as higher SOD activities than healthy controls (see *p*-values in Figure 1). In addition, PD patients presented higher GPx activity and lower GSH content (*p* < 0.05) than NDD-CKD patients. Interestingly, PD patients also exhibited lower and higher CAT and GPx activities compared with HD patients.

Finally, we also found a marked increase in lipid peroxidation in NDD-CKD, HD and PD patients (Figure 1F), showing higher MDA content (*p* < 0.001) than healthy subjects. Moreover, PD patients also showed higher MDA levels (*p* < 0.05) than NDD-CKD patients.

### 3.3. Changes in Oxidative Stress and Lipid Damage Parameters in Isolated Peripheral Blood Leukocytes

The results regarding oxidative stress and lipid damage parameters in isolated PMNs and MNs leukocytes from healthy subjects and NDD-CKD, HD and PD patients are shown in Figure 2 and Figure 3.

First, regarding both oxidant (XO) and antioxidant (CAT, GPx and SOD) enzymes (Figure 2), we found significant differences in PMNs and MNs from NDD-CKD patients relative to healthy controls. Thus, in both leukocytes populations, NDD-CKD patients exhibited higher XO activity (Figure 2A,B) and lower CAT (Figure 2C,D) and GPx (Figure 2E,F) activities than controls (see *p*-values in Figure 2). Interestingly, a different pattern was observed in the SOD activity, with NDD-CKD patients showing an increased SOD in PMNs (*p* < 0.05; Figure 2G) and a decreased SOD in MNs (*p* < 0.001; Figure 2H). Similar results were observed in HD and PD patients as compared with healthy controls. However, these redox state alterations were much more marked in PD patients, showing in both leukocytes population an increased XO activity and decreased CAT, GPx and SOD activities (see *p*-values in Figure 2). Furthermore, it highlights the redox state differences between NDD-CKD patients and dialysis patients. Thus, in both leukocytes populations, HD patients showed lower XO (Figure 2A,B) and CAT (Figure 2C,D) activities than NDD-CKD patients, also accompanied by decreased GPx activity in PMNs (Figure 2E). By contrast, an increased GPx and SOD activities (Figure 2F,H) were found in MNs from HD patients compared to NDD-CKD patients. Interestingly, in PD patients, these redox balance alterations were only observed in PMNs, showing higher CAT and lower SOD activities (Figure 2C,G) than NDD-CKD patients (see *p*-values in Figure 2). In regard to RRT, the PD group presented higher XO activity (Figure 2A,B) and lower SOD activity (Figure 2G,H) in PMNs and MNs compared with the HD group. In addition, PD patients also showed increased CAT and GPx activities in PMNs (Figure 2C,E), whereas in MNs, the GPx activity (Figure 2F) was significantly decreased in comparison to HD patients.

Secondly, regarding the glutathione balance, we observed significant differences in both GSH and GSSG contents and GSSG/GSH ratios in leukocytes from NDD-CKD, HD and PD patients relative to healthy controls (Figure 3). Interestingly, the MNs showed a greater alteration of glutathione than PMNs in dialysis patients. In general, NDD-CKD patients showed in both leukocytes populations significantly lower GSH contents (Figure 3A,B), as well as higher GSSG amounts (Figure 3C) and GSSG/GSH ratios (Figure 3E,F), than healthy subjects. However, HD and PD patients presented a marked increase of GSSG content and GSSG/GSH ratios in MNs (Figure 3D,F) compared with controls, also accompanied by a significant decrease of GSH content (Figure 3B) (see *p*-values in Figure 3). By contrast, no significant differences were observed in the glutathione parameters in PMNs between dialysis patients and healthy controls. In addition, HD and PD patients presented in PMNs a higher GSH content (Figure 3A) and lower GSSG levels and GSSG/GSH ratios (Figure 3C,E) than NDD-CKD patients. By contrast, these patients also presented a marked increase of GSSG content in MNs (Figure 3D) in comparison to NDD-CKD patients (see *p*-values in Figure 3). Regarding RRT, PD patients showed in MNs higher GSSG levels and GSSG/GSH ratio (*p* < 0.05; Figure 3D,F) compared with HD patients, whereas no differences were observed in PMNs.

Finally, regarding lipid damage (Figure 3G,H), NDD-CKD patients exhibited significantly higher MDA levels in PMNs and MNs (*p* < 0.01) than healthy subjects. Similar results were also observed in leukocytes from HD patients (MDA, *p* < 0.05) but not in PD patients. Interestingly, in both leukocyte populations, the PD group showed lower MDA content (*p* < 0.05*)* than NDD-CKD patients, whereas the HD group only showed this decreased MDA content in MNs (*p* < 0.01). Regarding RRT, PD patients had in PMNs significantly lower MDA content than HD patients (*p* < 0.05), whereas no differences were observed in MNs.

### 3.4. Correlations Between Biochemical, Inflammatory and Oxidative Stress Parameters in Plasma

The simple linear correlations between selected biochemical, inflammatory and oxidative stress parameters assessed in plasma from NDD-CKD, HD and PD patients are shown in Table 3.

In the NDD-CKD group, XO activity was positively associated with the inflammatory markers, such as IL-1β and CRP levels (*p* < 0.05). Remarkably, lipid damage (MDA content) was also positively related with the IL-1β release (*p* < 0.05) and HbA1c levels (*p* < 0.01). In addition, GPx activity was negatively associated with both SOD activity (*p* < 0.01), whereas a direct relationship was observed with both TG and LDL-C levels (*p* < 0.05). Interestingly, SOD activity was significantly negatively correlated with the LDL-C (p < 0.05), meanwhile GSH content directly correlated with the HDL-C (*p* < 0.05).

When the correlations were analyzed in the RRT groups, in HD patients, BMI was significantly positively correlated with the HbA1C. Interestingly, HbA1C content was also positively and negatively associated with the IL-1β and CAT (*p* < 0.05), respectively. Moreover, a positive correlation was also observed between XO and GPx activities (*p* < 0.01), and UA content and SOD activity (*p* < 0.05). Finally, the UA was positively correlated with the HDL-C (*p* < 0.05) in PD patients. No significant differences were found between the other variables analyzed (data not shown).

## 4. Discussion

To our knowledge, this is the first study that analyzed the changes in several redox state parameters in different types of isolated peripheral blood leukocytes, as well as in plasma, from advanced NDD-CKD patients, that also analyzed the effect of HD and PD procedures on redox status. Our results clearly revealed an altered redox state with a marked increase of oxidative stress and damage in plasma and isolated PMNs and MNs from NDD-CKD patients, and interestingly, is more exacerbated in HD and PD. Furthermore, we also found a different pattern of increased oxidative stress and damage depending on the localization (plasma or leukocytes) and cell type (PMNs and MNs). Thus, several of these redox state alterations were greatly impaired in PMNs, whereas others were only observed in MNs. Interestingly, our results also demonstrated differences between HD and PD procedures, the PD patients showing greater oxidative stress and damage than the HD patients, especially in MNs. This altered redox balance could be mediated by the higher inflammation and/or the changes in lymphocytes populations also observed in NDD-CKD, HD and PD patients.

A chronic oxidative-inflammatory state plays a crucial role in the onset and progression of CKD [5,12,13,14,15,16]. Indeed, numerous studies have demonstrated a significantly increased peripheral oxidative-inflammatory state in ESRD patients, being exacerbated by dialysis treatments [12,13,15,31,48]. Nevertheless, most research has been focused on the study of redox makers in serum or plasma, but not in immune cells. Since nutrition and diet significantly influence the extracellular redox status (plasma), and most enzymatic and non-enzymatic antioxidant mechanisms are intracellular [49], we analyzed the changes in several redox state parameters, not only in plasma but also in different types of peripheral leukocytes from advanced CKD patients. Furthermore, an optimal redox status of immune cells is essential for adequate homeostasis in the physiological systems [49]. In CKD patients, most studies have focused on lymphocyte populations, but little is known about the changes in the redox status of PMNs leukocytes along CKD progression. Since phagocytes have been proposed as the main cells that contribute to the oxidative stress associated with age-related diseases [9,10,11], we also assessed several redox markers in PMNs and MNs leukocytes in order to elucidate possible differences between them.

Regarding the plasma, our study demonstrated increased circulating levels of oxidative stress markers (higher XO activity and MDA content) and an impairment of the antioxidant systems (lower CAT and GPx activities and GSH concentration) in NDD-CKD, HD and PD patients, compared to healthy subjects. Our results are in consonance with other studies that have revealed increased oxidative stress markers associated with CKD [5,8,15,16,44]. This research demonstrated a high production of oxidant compounds (ROS, GSSG, etc.), lipid and protein oxidative damage markers (MDA, AOPPs, etc.), or altered levels of antioxidant (CAT, GPx, SOD, GSH, etc.) in plasma or serum of ESRD patients [5,8,15,16], as well as in HD and PD patients [12,13,14,15,28,31,32,48,50]. In this pathophysiological context, the possible malnutrition, dietary restrictions and the hypoalbuminemia state, together with the higher accumulation of non-dialyzable uremic toxins (e.g., indoxyl-sulphate, etc.) that CKD patients suffer, could contribute to the low availability of GSH and other antioxidants compounds and also promote higher oxidative-inflammatory stress in these subjects [5,48]. Indeed, the uremic toxins promote leukocyte activation, stimulating both ROS overproduction (mainly via XO and NADPH oxidase) and pro-inflammatory cytokines release [30,48]. The XO is a pro-oxidant enzyme involved in purine metabolism responsible for UA and ROS production [51]. On the one hand, circulating XO activity may be dangerous because once in circulation, it has the ability to activated phagocytic cells and produce O_2_^•−^ and H_2_O_2_, and it can be distributed to remote tissues [51,52], as well as internalized into vascular and other cells; therefore, it may further initiate oxidative damage exerting pathological effects in CKD patients [51,53]. On the other hand, although in optimal concentrations, UA has strong antioxidant properties, under conditions of oxidative stress, high UA concentrations may function as a powerful oxidizing compound, mainly when antioxidant defenses are diminished [8,27]. Moreover, hyperuricemia has been associated with CVDs and morbi-mortality in CKD patients, possibly through oxidative stress and endothelial dysfunction [48]. In our study, NDD-CKD patients showed higher UA levels than controls. Thus, these high UA concentrations, together with the increased XO activity and impaired antioxidant systems (GSH, GPx, CAT), could explain the higher plasma lipid damage (MDA content) also observed in NDD-CKD patients. MDA is a powerful oxidant compound, which can interact with others oxidized compounds, being potentially mutagenic and atherogenic [44]. Interestingly, it has been found that HD patients with CVDs had marked elevation of serum MDA compared with those without CVDs [33,54]. Moreover, Goundoin et al. found a positive correlation between plasma XO activity and MDA content in CKD and HD patients and proposed that, regardless of UA levels, plasma XO activity is an important predictor of CVDs in these patients [27]. Indeep, an increase in the circulating XO content could accelerate vascular oxidation via ROS generation, contributing to endothelial dysfunctions in CKD patients [55]. In our study, we also found higher percentages of CVDs in NDD-CKD and HD populations. However, the higher XO activity in plasma and isolated leukocytes were observed in NDD-CKD and PD patients, but not in HD patients, as will be discussed later.

Because oxidation and inflammation pathways can promote and exacerbate one another [9,10,11], we also analyzed the basal inflammation, measuring the IL-1β release and CRP levels in plasma, which provide higher specificity than other biomarkers (e.g., albumin) [2,56]. We observed increased CRP and IL-1β levels in NDD-CKD, HD and DP patients, which may contribute to triggering substantial collateral cellular oxidative damage in these patients. In fact, we also found a positive and significant association between IL-1β release and MDA content in NDD-CKD patients, as well as between IL-1β and HbA1c levels in HD patients. Although the studies are controversial, a strong upregulation of pro-inflammatory cytokines (e.g., IL-1β, IL-6, etc.) has been observed in CKD patients but also in dialyzed patients [5,15,50]. The increase of IL-1β has been also associated with the development of insulin resistance and modifications of lipid components, lipoproteins and proteins, resulting in lipid metabolism disorders in CKD [8]. In our study, a high percentage of NDD-CKD, HD and PD patients (79.5%, 59.0% and 63.6 %, respectively) suffered dyslipidemia, whereas NDD-CKD patients also showed marked hypertriglyceridemia. Because CRP is a powerful predictor of CVDs-morbidity and mortality in dialysis patients [6], the high plasma CRP levels observed in HD and PD patients may also contribute to CVDs in these patients. Moreover, because the outcome of local inflammation may directly build on the antioxidant ability [10], the impaired antioxidant systems observed in plasma from NDD-CKD, HD and PD patients may contribute to oxidative tissue damage, and consequently, to a systemic inflammatory response. Interestingly, although we did not find a relationship between inflammatory markers (IL-1β and CRP) and lipometabolism parameters in plasma from CKD patients, we observed a direct correlation between the GPx activity and both TG and LDL-C levels in NDD-CKD patients and between the GSH content and HLD-C levels. By contrast, SOD activity was significantly negatively correlated with LDL-C.

Regarding the isolated peripheral leukocytes, our study revealed enhanced oxidative stress and damage in PMNs and MNs from NDD-CKD, HD and PD patients, showing a marked increase of XO activity and lipid damage (MDA), an impairment of the antioxidant enzymes (higher SOD and lower CAT and GPx activities) and an altered redox GSSG/GSH balance compared to healthy controls. Interestingly, we observed notable differences in the redox status between PMNs and MNs in NDD-CKD, as well as between HD and PD patients. Concerning the pro-oxidant XO enzyme, our results revealed a higher XO activity in PMNs and MNs from NDD-CKD and PD patients, which can promote ROS overproduction, as well as exert cytotoxicity on the kidney and the vascular endothelium during CKD progression [16]. Since XO and its products are directly and indirectly implicated in immunomodulatory activities [52], this excessive increase in XO activity could induce the activation of PMNs, promoting phagocytic killing, the expression of other pro-oxidant enzymes (NADPH oxidase, etc.) or the nuclear factor-κB activation and translocation, and consequently, a pro-inflammatory cytokines overproduction [51,52,53]. Moreover, because the XO gene expression is markedly upregulated by pro-inflammatory cytokines (e.g., IL-1β) [51], the higher inflammation observed in NDD-CKD patients could also mediate a higher expression of XO, explaining the increased XO activity and the positive correlation observed between XO activity and inflammatory markers (IL-1β and CRP levels) in these patients. Interestingly, HD patients showed a marked reduction of XO activity in plasma, PMNs and MNs compared to PD and NDD-CKD patients. Although the research regarding XO in dialyzed patients is not clear, one study revealed that XO activity was markedly elevated in HD and PD patients independently of dialysis modality [57], whereas other authors proposed that the extracellular XO activity differ depending on the nutritional status of the HD patients [53]. Indeep, serum XO activity decreased during HD procedure in patients with high nutritional risk index and vice versa [53]. Although the decreased XO activity in HD patients appears a priori to be beneficial for these subjects, it could also be indicative of a basically higher content of endothelium-bound XO. Further, circulating XO has the ability to reversibly bind to the endothelial cell surface via glycosaminoglycan rich receptors, leading to ROS overproduction [51,52,53], especially at locations exposed to mechanical forces [53,58]. Thus, we propose that HD patients could have high circulating XO binding to the extracellular matrix of endothelial cells. This can promote increased lipid damage and the inflammatory cascade activation, and therefore, cause vascular dysfunction in HD patients [53]. In fact, increased XO levels have been documented in the arteries of HD patients [53]. Further studies are needed to elucidate the endothelium-bound XO and its effects on oxidative damage in chronic HD and PD patients.

Concerning the SOD, CAT and GPx antioxidant enzymes, which are essential to prevent oxidative damage throughout CKD progression [5,16], we found striking differences. The SOD catalyzes the conversion of O_2_^•−^ into H_2_O_2_ and oxygen, while CAT and GPx mediate the conversion of toxic H_2_O_2_ into water. Under normal conditions, H_2_O_2_ is principally degraded by GPx (due to its higher affinity for H_2_O_2_ than CAT), whereas under severe oxidative stress conditions, H_2_O_2_ is mainly degraded by CAT [49]. On the one hand, NDD-CKD patients showed lower SOD activity in MNs and higher SOD in PMNs and plasma in comparison with healthy subjects. This increased SOD activity may be a balancing mechanism to neutralize the excessive O_2_^•−^ levels, as has also been observed in both plasma and PMNs of CKD patients [8,11,12,13]. Paradoxically, PD patients had a decreased SOD activity in PMNs compared with NDD-CKD patients, whereas HD patients presented an enhanced SOD activity in MNs. Most studies have shown higher SOD activity in the serum of HD patients, being associated with the impaired renal functions [16]. Other studies have found high SOD gene expression in leukocytes from ESRD patients, which also correlated with high plasma CRP and O_2_^•−^ levels [6,59]. Interestingly, we found a positive correlation between SOD activity and UA content in HD patients. On the other hand, we also observed a significant down-regulation of CAT and GPx in PMNs and MNs from NDD-CKD, HD and PD patients. This is in agreement with several studies that have shown a gradual decrease in the plasma GPx and CAT activities along with the CKD progression [8,12,13,16]. Interestingly, HD patients presented the most marked alteration of these antioxidant enzymes, especially in PMNs. Indeep, HD patients showed lower CAT and GPx activities in PMNs than NDD-CKD patients, whereas increased GPx activity was found in MNs, which could be a compensatory mechanism to protect against excessive oxidative stress. In fact, an increased GPx activity in erythrocytes protects against Hb oxidation and hemolysis, preventing against CVDs in CKD patients [16]. Interestingly, we found a negative correlation between GPx and SOD activities in NDD-CKD patients. Meanwhile, in HD patients, GPx activity was positively associated with XO activity, whereas CAT activity was negatively correlated with HbA1c content. Furthermore, an excess of SOD, CAT and GPx activities may promote detrimental outcomes as a consequence of a deficit of vital cell oxidants (optimal ROS concentrations), causing “reductive stress,” which could contribute to weakening cell dysfunctions [16,49]. GPx not only detoxifies H_2_O_2_ but also catalyzes the reduction of lipid peroxides, thus preventing lipid peroxidation [49]. Therefore, the altered GPx activity observed in leukocytes from NDD-CKD patients may promote the accumulation of lipoperoxides, explaining the higher MDA content also observed in both PMNs and MNs of these patients. Our results are in agreement with other studies that also found elevated MDA levels in CKD patients undergoing HD [25,54]. MDA is a low molecular weight and water-soluble molecule, so it can undergo renal clearance or be dialyzed as a possible renal elimination pathway [25]. Interestingly, HD and PD patients had lower MDA content in their leukocytes than NDD-CKD patients. Although we have not found any study that has evaluated the differences in lipid peroxidation between PMNs and MNs in both NDD-CKD and dialysis patients, it has been proposed that MDA markedly decreased during the HD procedure [12,13,14,31]. Due to the absence of blood contact with artificial surfaces, PD patients are a particularly interesting population to study the effect of uremia on lipid peroxidation [12,31,38]. Interestingly, in our study, PD patients showed lower MDA levels in their leukocytes compared to HD patients. In this context, it has been described that PD promotes protein oxidation, whereas HD more intensely impacts lipid peroxidation [50].

The glutathione cycle is one of the main intracellular mechanisms to preserve an adequate intracellular redox state [11,60]. In fact, GSH is the major antioxidant involving in cell-signaling regulation, whereas GSSG is highly toxic to cells [60]. Therefore, an optimal GSSG/GSH balance is essential for the preservation of oxidative damage and the maintenance of immune functions [11,60]. In our study, NDD-CKD patients showed in PMNs and MNs lower GSH content, together with higher GSSG and GSSG/GSH ratios, compared with controls. Interestingly, this altered GSSG/GSH balance was more exacerbated in PMNs than in MNs leukocytes. Since GSH is essential to neutralize ROS, its reduction could contribute to higher accumulation of O_2_^•−^ and H_2_O_2_ (mainly via XO or NADPH oxidase production), and together with the increased GSSG content, could explain the marked increase of lipid damage (MDA) also observed in PMNs from NDD-CKD patients. Interestingly, RRT had a different influence on the GSSG/GSH balance depending on the cell typed analyzed. Moreover, HD and PD procedures had a positive effect on the GSSG/GSH status in PMNs, with dialysis patients showing higher GSH content and lower GSSG levels and GSSG/GSH ratio than NDD-CKD patients. This could be a compensatory response in PMNs to cope with high exposure to oxidant compounds. By contrast, lower GSH and higher GSSG content and GSSG/GSH ratios were observed in MNs from HD and PD patients compared with NDD-CKD patients. These findings are in line with previous research showing decreased GSH and increased GSSG and GSSG/GSH in lymphocytes or whole-blood from HD or PD patients [61,62].

In regards to RRT, it is known that HD and PD patients presented both ROS overproduction and mostly compromised antioxidant mechanisms [12,13,14,31,38]. In fact, HD therapy per se seems to contribute to the oxidant/antioxidant imbalance due to several factors (e.g., the biocompatibility of dialysis membrane and dialysates, etc.) [8,14,31,38], whereas PD does not seem to have a strong benefit over HD procedure, even though the biocompatibility advantage [38]. Indeed, several studies have indicated that increased ROS production in peripheral/peritoneal phagocytes is activated during PD, mainly due to the composition of PD dialysates [12,32]. Interestingly, in our group of PD patients, we found oxidative stress values significantly higher in plasma and isolated leukocytes compared to HD patients, especially in MNs. Thus, PD patients showed a higher XO activity and lower SOD activity in PMNs and MNs than HD patients. However, in MNs, PD patients had a significant reduction of SOD and GPx activities, together with increased GSSG and GSSG/GSH ratio, which may contribute to the accumulation of ROS and lipid damage in these patients. By contrast, PD patients had in PMNs higher CAT and GPx activities and a preserved GSSG/GSH ratio compared with HD patients, suggesting that these compensatory antioxidant mechanisms could contribute to the reduced MDA content also observed in these cells. Overall, based on the limited data reviewed so far, some studies (mainly in plasma) showed increased oxidative stress and damage in HD compared with PD, whereas others reported no changes [12,13,14,31,32,33,38,44]. By contrast, some authors proposed that PD is associated with a higher accumulation of oxidant compounds and antioxidant depletion [38,57]. Furthermore, a preserved RRF has been related to reduced oxidative stress and lipid damage in PD patients [12,42,43]. However, although PD patients showed higher RRF than HD patients, we did not observe significant correlations between RRF and oxidative-inflammatory parameters analyzed (data not shown) in these patients. Based on our results, we deduce that PD procedure may contribute to increased oxidative stress on CKD patients, and especially in MNs leukocytes.

An optimal redox state is essential for the effectiveness of the immune functions, which can also be influenced by the changes in the leukocyte populations in terms of the cell types composition, quality and proportions [9,10]. Much evidence indicates that lymphocytes impairment is also involved in immune dysregulation in ESRD patients [46,63,64]. We found remarkable changes with respect to MNs populations (mainly lymphocytes), but not in PMNs (>98% were neutrophils in all groups; data not shown). In several studies, lymphocyte subgroups were lower in dialysis patients than in healthy controls [63,64]. In our study, T- and B-cells were also influenced by dialysis modality. Thus, HD and PD patients showed a lower total number of lymphocytes and CD4+/CD8+ ratio than NDD-CKD patients. Interestingly, HD patients also had a marked reduction of CD4+ T-lymphocytes and NK cells than NDD-CKD patients, whereas PD patients had lower B-cell counts. Our results are in agreement with other studies in which dialysis patients exhibited a high incidence of lymphopenia affecting T- and B-cell populations [46,63,64]. The decreased CD4+ T-cells proportion could promote infection and mortality in HD patients [63]. Moreover, the differences in both uremic toxin accumulation and inflammatory status between HD and PD patients could play a key role in the distribution of T- and B-lymphocytes on these subjects [63,65]. Indeed, HD is associated with more sustained inflammation and lymphocyte activation/exhaustion [65], explaining the results observed in HD patients. Because of the cross-sectional nature of our study, we have to confirm these changes in a very large cohort as well as investigate modifications in immunological parameters after and during HD and PD procedures.

Finally, our study has several limitations: (1) the small patient´s sample size, especially in the PD group, which may cause not to find significant differences in several of the parameters analyzed. (2) Our study did not interfere with or evaluate the lifestyle factors of the CKD patients (nutritional status, diet, etc.) and other non-conventional risks (e.g., anemia, etc.), which also contribute to oxidative stress and inflammation. (3) The cross-sectional design and analysis of data, and the small number of subjects, did not allow us to integrate and analyze the influences of all potential covariables (age, gender, type of dialysates, the different dialysis membranes, vascular access, comorbidities, etc.) into the multivariate models. Therefore, additional prospective and comparative clinical studies in a larger sample will be required to validate our findings and confirm the use of these redox state parameters as biomarkers of CKD progression. Moreover, future large studies addressing potential nutritional interventions to decrease oxidative-inflammatory stress in leukocytes from CKD patients are also necessary.

## 5. Conclusions

The major strength of this study is that we analyzed the changes of numerous redox state markers in both plasma and isolated peripheral blood leukocytes from advanced CKD patients and also analyzed the effect of RRT on redox status. Taken together, our study demonstrated that NDD-CKD, HD and PD patients presented significantly higher oxidative stress and damage in their PMNs and MNs leukocytes than healthy subjects, as well as in plasma. Interestingly, the oxidative stress and damage were more exacerbated in HD and PD patients than in NDD-CKD patients. This enhanced oxidative stress could be mediated by a higher inflammation state, and the changes in lymphocytes populations were also observed in these patients. Interestingly, we also found notable differences in the redox state depending on the localization (plasma/leukocytes) and the immune cell type (PMNs/MNs) analyzed. In fact, several redox state alterations were greatly marked in plasma or PMNs, whereas others were only observed in MNs. Thus, dialysis treatments had a positive effect preserving the GSSG/GSH balance in PMNs, but not in MNs. This study also demonstrated remarkable differences between HD and PD procedures, the PD patients showing greater oxidative stress and damage than the HD patients, especially in MNs. Our results could explain one of the underlying mechanisms of immune dysregulation in CKD patients. Furthermore, since PMNs and MNs are very easy to obtain and due to the simplicity of the assays performed, the assessment of several redox state parameters in these leukocytes, together with the plasma, could be used as potential biomarkers in the CKD progression, as well as in monitoring HD and PD procedures in a clinical setting.

## Figures and Tables

**Figure 1 antioxidants-10-01155-f001:**
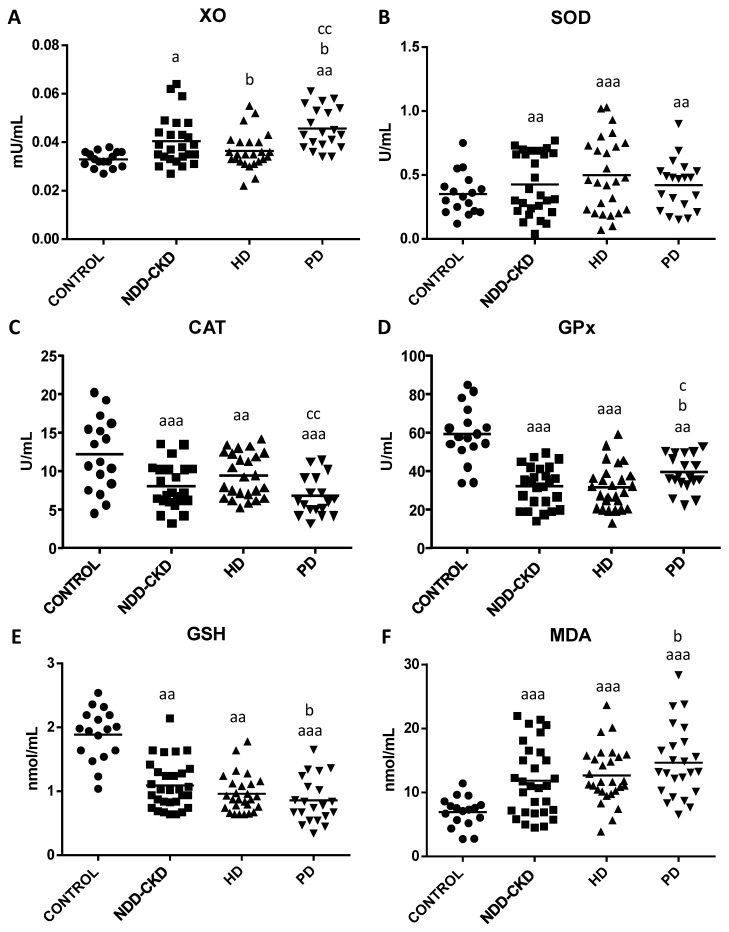
Oxidative stress and lipid damage parameters in plasma from healthy subjects (control), non-dialysis-dependent chronic kidney disease (NDD-CKD), hemodialysis (HD) and peritoneal dialysis (PD) patients. (**A**) Xanthine oxidase (XO; mU/mL), (**B**) superoxide dismutase (SOD; U/mL), (**C**) catalase (CAT; U/mL) and (**D**) glutathione peroxidase (GPx; U/mL) activities; (**E**) reduced glutathione (GSH; nmol/mL) and (**F**) malondialdehyde (MDA; nmol/mL) contents. Data are shown as the mean (horizontal bar) of 17–39 values corresponding to the number of subjects analyzed in each group. Each value is the mean of duplicate assays. ^a^ *p* < 0.05, ^aa^ *p* < 0.01 and ^aaa^ *p* < 0.001 versus control; ^b^ *p* < 0.05 versus NDD-CKD patients; ^c^ *p* < 0.05 and ^cc^ *p* < 0.01 versus HD patients.

**Figure 2 antioxidants-10-01155-f002:**
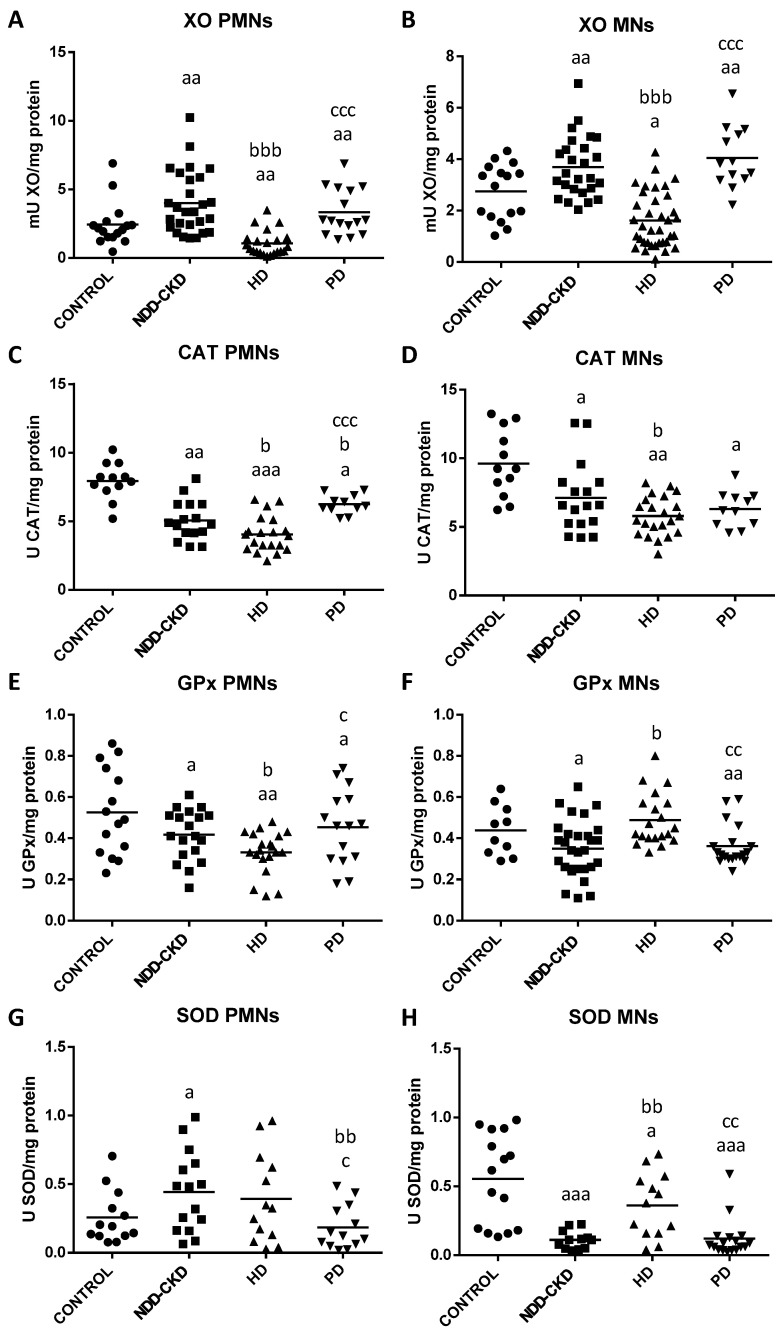
Oxidative stress parameters in isolated peripheral blood polymorphonuclear (PMNs) and mononuclear (MNs) leukocytes from healthy subjects (control), non-dialysis-dependent chronic kidney disease (NDD-CKD), hemodialysis (HD) and peritoneal dialysis (PD) patients. (**A**) PMNs and (**B**) MNs xanthine oxidase activity (XO; mU/mg protein); (**C**) PMNs and (**D**) MNs catalase activity (CAT; U/mg protein); (**E**) PMNs and (**F**) MNs glutathione peroxidase activity (GPx; U/mg protein); (**G**) PMNs and (**H**) MNs superoxide dismutase activity (SOD; U/mg protein). Data are shown as the mean (horizontal bar) of 10–32 values corresponding to the number of subjects analyzed in each group. Each value is the mean of duplicate assays. ^a^
*p* < 0.05, ^aa^ *p* < 0.01 and ^aaa^ *p* < 0.001 versus control; ^b^ *p* < 0.05, ^bb^ *p* < 0.01 and ^bbb^ *p* < 0.001 versus NDD-CKD patients; ^c^ *p* < 0.05, ^cc^ *p* < 0.01 and ^ccc^ *p* < 0.001 versus HD patients.

**Figure 3 antioxidants-10-01155-f003:**
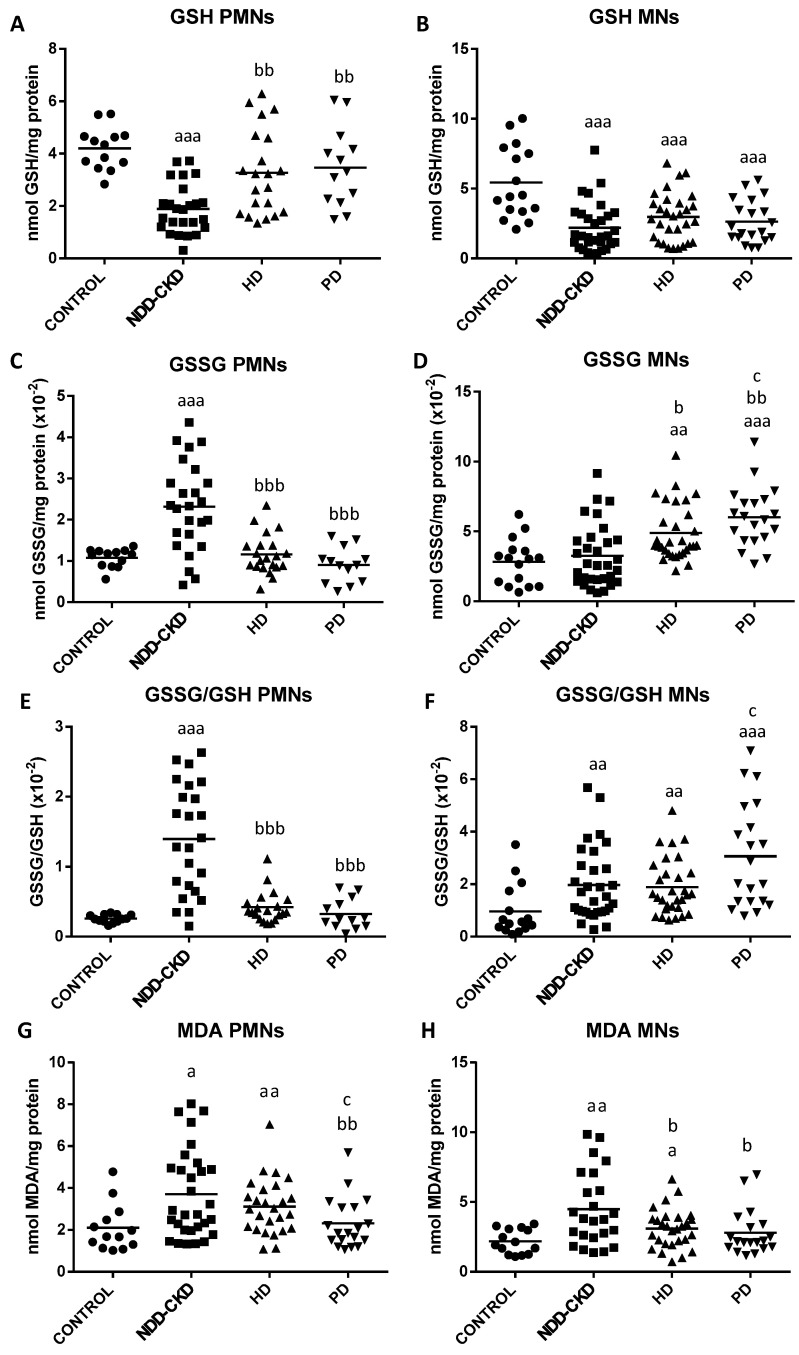
Intracellular glutathione balance and lipid damage in isolated peripheral blood polymorphonuclear (PMNs) and mononuclear (MNs) leukocytes from healthy subjects (control), non-dialysis-dependent chronic kidney disease (NDD-CKD), hemodialysis (HD) and peritoneal dialysis (PD) patients. (**A**) PMNs and (**B**) MNs reduced glutathione content (GSH; nmol/mg protein); (**C**) PMNs and (**D**) MNs oxidized glutathione content (GSSG; nmol/mg protein); (**E**) PMNs and (**F**) MNs GSSG/GSH ratio. (**G**) PMNs and (**H**) MNs malondialdehyde content (MDA; nmol/mg protein). Data are shown as the mean (horizontal bar) of 13–39 values corresponding to the number of subjects analyzed in each group. Each value is the mean of duplicate assays. ^a^ *p* < 0.05, ^aa^
*p* < 0.01 and ^aaa^ *p* < 0.001 versus control; ^b^ *p* < 0.05, ^bb^ *p* < 0.01 and ^bbb^ *p* < 0.001 versus NDD-CKD patients; ^c^ *p* < 0.05 versus HD patients.

**Table 1 antioxidants-10-01155-t001:** The demographic and clinical characteristics of the study population.

Characteristics	Control	NDD-CKD	HD	PD
Demographic data	*n* = 17	*n* = 39	*n =* 39	*n =* 22
Male, *n* (%)	8 (47.1)	25 (64.1)	27 (69.2)	11 (50)
Female, *n* (%)	9 (52.9)	14 (35.9)	12 (30.8)	11 (50)
Age (years; mean ± SD)	52.17 ± 15.33	60.41 ± 17.26	57.46 ± 14.75	54.63 ± 15.82
BMI (kg/m^2^; mean ± SD)	24.54 ± 3.55	25.19 ± 9.17	24.29 ± 4.05	24.54 ± 4.17
eGFR (mL/min per 1.73 m^2^)	-	15.84 ± 2.74	-	-
Comorbidity				
Hypertension, *n* (%)	1 (5.9)	35 (89.7)	33 (84.6)	20 (90.9)
Diabetes, *n* (%)	2 (11.8)	17 (43.6)	7 (17.9) ^bb^	5 (22.7) ^b^
Dyslipidemia, *n* (%)	0 (0)	31 (79.5)	23 (59.0) ^b^	14 (63.6) ^b^
Hyperuricemia, *n* (%)	0 (0)	27 (69.2)	10 (25.6) ^bb^	13 (59.1) ^c^
CVD, *n* (%)	0 (0)	19 (48.7)	19 (48.7)	7 (31.8)
Peripheral vascular disease, *n* (%)	0 (0)	1 (2.6)	9 (23.1) ^b^	2 (9.1) ^c^
ACVA, *n* (%)	0 (0)	5 (12.8)	2 (5.1)	3 (13.6)
Tumors, *n* (%)	0 (0)	1 (2.6)	1 (2.6)	1 (4.5)
Etiology of the nephropathy				
Glomerulonephritis, *n* (%)	0 (0)	6 (15)	11 (28)	8 (36)
Hypertension/vascular, *n* (%)	0 (0)	7 (18)	6 (15)	4 (18)
Diabetes, *n* (%)	0 (0)	12 (31)	4 (10)	4 (18)
Polycystic kidneys	0 (0)	4 (10)	1 (3)	1 (5)
Tubulointersticial nephritis, *n* (%)	0 (0)	6 (15)	8 (21)	2 (5)
Others, *n* (%)	0 (0)	4 (10)	9 (23)	3 (14)
Dialysis data				
AVF (%)	-	-	56.41	-
OL-HDF/HFD (%)	-	-	75/25	-
APD/CAPD (%)	-	-	-	86/14
Kt/v (mean ± SD)	-	-	1.67 ± 0.25	2.34 ± 0.51
RRF (mL/day; median)	-	-	400 (100–1300)	1275 (500–3000) ^cc^
Treatment				
Statins	0 (0)	30 (76.9)	16 (41) ^b^	14 (63.6) ^c^
Allopurinol	0 (0)	23 (59)	8 (20.5) ^b^	14 (63.6) ^c^
Antiplatelet	0 (0)	8 (20.5)	16 (41) ^b^	7 (31.8)
Erythropoietin	0 (0)	18 (46.2)	34 (87.2) ^bb^	16 (72.7) ^bb^

Abbreviations: ACVA, cerebral vascular accident; APD, automated peritoneal dialysis; AVF, arteriovenous fistula; BMI, body mass index; CAPD, continuous ambulatory peritoneal dialysis; CVD, cardiovascular disease; eGFR, estimated glomerular filtration rate; HD, hemodialysis; HFD, high-flux dialysis; Kt/v, standardized dialysis adequacy. NDD-CKD, non-dialysis-dependent chronic kidney disease; OL-HDF, online hemodiafiltration; PD, peritoneal dialysis; SD, standard deviation. ^b^ *p* < 0.05 and ^bb^ *p* < 0.01 versus NDD-CKD patients. ^c^ *p* < 0.05 and ^cc^ *p* < 0.01 versus HD patients. (-): data not included.

**Table 2 antioxidants-10-01155-t002:** Biochemical and inflammatory parameters and lymphocytes population of the study participants.

Variables	CONTROL	NDD-CKD	HD	PD
Biochemical parameters	*n =* 17	*n* = 39	*n* = 39	*n* = 22
Serum creatinine (mg/dL)	0.82 ± 0.16	4.17 ± 1.04 ^aaa^	7.79 ± 1.91 ^aaa,bbb^	7.46 ± 2.72 ^aaa,bbb^
Albumin (g/dL)	4.65 ± 0.28	4.25 ± 0.36 ^aa^	4.11 ± 0.35 ^aa^	3.82 ± 0.43 ^aaa,bb^
TG (mmol/L)	96.94 ± 43.44	171.02 ± 98.36 ^aa^	117.23 ± 44.70 ^bb^	132.68 ± 61.64 ^b^
TC (mmol/L)	178.41 ± 27.64	168.41 ± 52.30	147.15 ± 30.96 ^b^	159.45 ± 37.39
HDL-C (mmol/L)	55.17 ± 8.88	46.02 ± 13.43	48.72 ± 11.21	47.41 ± 13.80
LDL-C (mmol/L)	103.82 ± 26.33	89.85 ± 44.32	77.82 ± 31.30	90.45 ± 28.03
UA (mg/dL)	5.10 ± 1.10	6.32 ± 1.70 ^a^	5.78 ± 1.22	5.76 ± 1.29
Hb (g/dL)	13.61 ± 2.30	9.98 ± 2.82 ^a^	8.79 ± 2.85 ^aa^	10.21 ± 2.52 ^a^
HbA1c (mg/dL)	5.43 ± 0.43	5.95 ± 1.20	5.14 ± 0.70 ^b^	42 ± 0.49
Inflammatory markers				
CRP (mg/dL)	0.29 ± 0.09	0.45 ± 0.07 ^a^	0.86 ± 0.13 ^aa,b^	0.83 ± 0.16 ^aa,b^
IL-1β (pg/mL)	4247 ± 1218	5200 ± 2450 ^a^	5156 ± 2697	5280 ± 3031
Lymphocyte populations (cells/µL)				
Total lymphocytes	1798 ± 449	1585 ± 548	1083 ± 509 ^aaa,bbb^	1271 ± 293 ^aa,b^
CD3^+^ T-cells	1272 ± 350	1187 ± 440	761 ± 425 ^aaa,bbb^	916 ± 537 ^a^
CD4^+^ T-cells	831 ± 308	731 ± 295	430 ± 280 ^aaa,bbb^	603 ± 224 ^a^
CD8^+^ T-cells	410 ± 143	429 ± 250	301 ± 190	387 ± 136
CD4^+^/CD8^+^ ratio	2.23 ± 0.22	2.20 ± 0.17	1.49 ± 0.09 ^a,bbb^	1.71 ± 0.21 ^b^
CD19^+^ B-cells	194 ± 22	130 ± 18	118 ± 21 ^a^	81 ± 11 ^aa,b^
CD3^−^CD56^+^CD16^+^ NK	243 ± 23	204 ± 22	173 ± 24 ^a,b^	228 ± 13

Abbreviations: CRP, C-reactive protein; Hb, hemoglobin; HbA1c, glycosylated hemoglobin; HDL-C, high-density lipoprotein cholesterol; HD, hemodialysis; IL-1β, interleukine-1β; LDL-C, low-density lipoprotein cholesterol; NK, natural killer cells; NDD-CKD, non-dialysis-dependent chronic kidney disease; PD, peritoneal dialysis; TC, total cholesterol; TG, triglycerides; UA, uric acid. Data are shown as mean ± standard deviation. ^a^ *p* < 0.05, ^aa^ *p <* 0.01 and ^aaa^ *p* < 0.001 versus control; ^b^ *p* < 0.05, ^bb^ *p* < 0.01 and ^bbb^ *p* < 0.001 versus NDD-CKD patients.

**Table 3 antioxidants-10-01155-t003:** Pearson´s correlation coefficients (*r*) between selected biochemical, inflammatory and oxidative stress parameters in plasma of study participants.

Group	Parameters	*r*	*p*-Value
NDD-CKD	XO; IL-1β	0.333	0.048 *
	XO; CRP	0.776	0.039 *
	IL-1β; MDA	0.511	0.036 *
	SOD; GPx	−0.579	0.002 **
	SOD; LDL-C	−0.689	0.040 *
	GPx; TG	0.810	0.020 *
	GPx; LDL-C	0.775	0.041 *
	GSH; HDL-C	0.651	0.022 *
	MDA; HbA1c	0.676	0.010 **
HD	BMI; HbA1c	0.357	0.028 *
	IL-1β; HbA1c	0.553	0.006 **
	XO; GPx	0.598	0.003 **
	CAT; HbA1c	−0.434	0.043 *
	UA; SOD	0.478	0.022 *
PD	UA; HDL-C	−0.451	0.035 *

Abbreviations: BMI, body mass index; CAT, catalase activity; CRP, C-reactive protein; GPx, glutathione peroxidase activity; GSH, reduced glutathione levels; HbA1c, glycosylated hemoglobin; HDL-C, high-density lipoprotein cholesterol; HD, hemodialysis; IL-1β, interleukine-1β release; LDL-C, low-density lipoprotein cholesterol; MDA, malondialdehyde levels; NDD-CKD, non-dialysis-dependent chronic kidney disease; PD, peritoneal dialysis; SOD, superoxide dismutase activity; TG, triglycerides; UA, uric acid; XO, xanthine oxidase activity. * *p* < 0.05 and ** *p* < 0.01.

## Data Availability

Data is contained within the article.

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
