# Peer review of "Oxidative Stress in Patients with Advanced CKD and Renal Replacement Therapy: The Key Role of Peripheral Blood Leukocytes"

_antioxidants, 2021, doi:10.3390/antiox10071155_

Round 1
Reviewer 1 Report
Dear Author,
I read this paper, in which the authors report the results of a study exploring the redox status in plasma and peripheral leukocytes in patients with different stages of kidney damage.
In general, I have found the paper interesting for its specific topic.
However, I think that some clarifications are needed.
SPECIFIC COMMENTS:
Patients and Methods:
- What was the timing of the sample collection in HD patients?
- What dialysis membranes were used?
- Please indicate if any patients presented residual renal function (to add to Table 1)
- Among the lab parameters in the different groups, I suggest also considering Hb levels (to add to Table 2).
Please, notice and discuss that both residual renal function and anemia can impact redox status in patients with kidney diseases
(for example, see Ignace S, et al. Preserved residual renal function is associated with lower oxidative stress in peritoneal dialysis patients. Nephrol Dial Transplant. 2009 May;24(5):1685-9 and Grune T, et al.. Oxidative stress in anemia. Clin Nephrol. 2000 Feb;53(1 Suppl):S18-22)
- What was the eGFR value of ESRD patients. Were these patients on a low protein diet?
RESULTS
-Please, add significance in Table 1
- The descriptions of the results on the analysis of redox markers can be hard to read (especially the first one about the circulating markers). I advise reformulating and simplifying these paragraphs (also considering that some data, as "p", are also reported in the figures)
- There were correlations among UA, inflammatory parameters and redox markers?
DISCUSSION
- Apart from the considerations on renal residual function and anemia, I think that it should be emphasized the impact of diet on redox status in renal patients (for example, see Garibotto G, et al. Muscle protein turnover and low-protein diets in patients with chronic kidney disease. Nephrol Dial Transplant. 2020 May 1;35(5):741-751).
- Moreover, it should be mentioned that there is evidence on the role of different dialysis fluids in PD and membranes in HD as potential modulators of redox in dialysis patients (see, Sepe V, et al. Vitamin e-loaded membrane dialyzers reduce hemodialysis inflammaging. BMC Nephrol. 2019 Nov 15;20(1):412.)
I think that the absence of a specific assessment of these aspects (mainly because of the small patient sample and cross-sectional design) could constitute a limit of this study, but could also provide suggestions and opportunities to designed further researches. Please, discuss this point.
Author Response
Response to the reviewer´s comment. "Please see the attachment"

Reviewer 2 Report
The paper is very interesting and well written. The length of the paper seems excessive to me with a possible effect on the reading, especially as regards the materials and methods.
Regarding the treated topic I have some questions:
What role play the dialysis membrane and the glucose concentration in the peritoneal dialysis solution in the final oxidative stress?In the detail for HD: the type of membrane, the type of dialysate and reinfusion fluids (lactate/acetate/citrate/HCl buffer)? The anticoagulation with heparin is known to increase the inflammatory state of the patient: was heparin or low molecular weight heparin used in HD treatments? From table 2 it seems that all patients had HD, is it correct? No HDF or AFB o HFR? I suggest to introduce in the paper some bibliography about the inflammation / oxidative stress according to the different KRTs.
The glucose concentrations of the PD solutions and the type of PD have not been described.
I suggest the authors to cut some of the materials and methods to better explain these points.
Author Response
Response to the reviewer´s comments. "Please see the attachment"

Reviewer 3 Report
In their clinical study, Vida et al. investigate different markers of oxidative stress in plasma and isolated peripheral polymorphonuclear (PMNs) and mononuclear (MNs) from healthy patients, patients with End-Stage-Renal-Disease (ESRD) and renal replacement therapy (hemodialysis (HD) and peritoneal dialysis (PD)). The authors find increased markers of oxidative stress in both plasma and leukocytes in patients with kidney disease compared to healthy controls.
The study design and the biochemical analyses are straight forward and solid. However, the results are only descriptive and there are no correlations or other outcome-related analyses of the results.
- In the Method part it is stated that 9 NDD-ESRD patients were included but in the tables there are n=39. Please clarify.
- When was the study performed? Please mention the year or time period.
- Please rename the ESRD-group as NDD-CKD and add the eGFR of this particular group in Table 1
- Did the authors perform ANCOVA to rule out influences of covariates such as sex, age, medication and others on the results? Was the covariate creatinine the only one that was significantly associated with the differences between the groups?
- I think that plotting the results of XO and the other biochemical markers measured in the different compartments (plasma, PMNs, MNs) into one line would be better to see and compare the pattern of each parameter (e.g. XO plasma, XO PMNS and XO MN and so on). This could be done line for line. The results could be distributed in 2 figures. In the results the description should be adapted accordingly.
- The authors measured several lipometabolism-parameters in the blood of the patients. Especially in Table 2 it would be very interesting to add the BMI and see the correlation between the lipometabolism and the individual BMI. It is already known that obesity leads to increased oxidative stress. It would be very interesting to check whether there is a correlation between the individual groups studied, the BMI and the markers of oxidative stress collected.
- The legend in figure two appears to have a content error or is not clear. From line 351 it reads: “T1< p < 0.05 significant statistical trend with respect control.”
However, from our point of view it should read as follows: “T0.05< p < 0.1 significant statistical trend with respect control.” Please clarify.
- In their summary, the authors state that the assessment of redox state parameters in PMNs and MNs could have potential use as biomarkers of the ESRD progression. How might this affect every day clinical practice? What action-specific consequences could result from the knowledge gained, in particular the differences between HD and PD patients?
Author Response

(The authors gave the same response as above.)

Round 2
Reviewer 1 Report
Thank you to the authors for the full revision of their paper, which I think is significantly improved in the current version.
Reviewer 3 Report
The authors have responded to my comments and I recommend its acceptance.